# Vibration Control of Flexible Launch Vehicles Using Fiber Bragg Grating Sensor Arrays

**DOI:** 10.3390/s25010204

**Published:** 2025-01-02

**Authors:** Bartel van der Veek, Hector Gutierrez, Brian Wise, Daniel Kirk, Leon van Barschot

**Affiliations:** 1Department of Electrical and Computer Engineering, Florida Institute of Technology, Melbourne, FL 32901, USA; bartel.vanderveek@gmail.com; 2Department of Mechanical Engineering, Florida Institute of Technology, Melbourne, FL 32901, USA; brian.wise@genevatech.net; 3Department of Aerospace Engineering, Physics and Space Science, Florida Institute of Technology, Melbourne, FL 32901, USA; dkirk@fit.edu; 4Department of Electrical Engineering, Technical University Eindhoven, 5612 AZ Eindhoven, The Netherlands; leonvanbarschot@gmail.com

**Keywords:** fiber Bragg sensor array, strain sensor, flexible structure, modal shapes, real time control

## Abstract

The effects of mechanical vibrations on control system stability could be significant in control systems designed on the assumption of rigid-body dynamics, such as launch vehicles. Vibrational loads can also cause damage to launch vehicles due to fatigue or excitation of structural resonances. This paper investigates a method to control structural vibrations in real time using a finite number of strain measurements from a fiber Bragg grating (FBG) sensor array. A scaled test article representative of the structural dynamics associated with an actual launch vehicle was designed and built. The main modal frequencies of the test specimen are extracted from finite element analysis. A model of the test article is developed, including frequency response, thruster dynamics, and sensor conversion matrices. A model-based robust controller is presented to minimize vibrations in the test article by using FBG measurements to calculate the required thrust in two cold gas actuators. Controller performance is validated both in simulation and on experiments with the proposed test article. The proposed controller achieves a 94% reduction in peak–peak vibration in the first mode, and 80% reduction in peak–peak vibration in the second mode, compared to the open loop response under continuously excited base motion.

## 1. Introduction

Modern space launch vehicles (LVs) tend towards smaller weight and lower aerodynamic drag: designs are becoming more slender, more flexible and prone to structural vibrations [1]. The effect of vibrations on control system stability could be dominant in systems designed assuming rigid-body dynamics, since it acts as unmodeled dynamics [2,3]. Current control methods for flexible LVs are mostly focused on preventing bending vibration signals from feeding back to the trajectory control system [2,3,4,5,6,7,8,9,10]. Direct vibration suppression (suppression based on actuation) is not currently used in LV control, partly because conventional sensor systems for vibration estimation require substantial wiring at multiple locations within a LV, which makes them cumbersome, susceptible to electromagnetic interference, and are burdened by a significant weight penalty due to the required wiring. Fiber Bragg grating (FBG) sensors [11,12,13,14] are small, lightweight, easy to multiplex, immune to electromagnetic interference, and easy to install in host structures. These advantages, combined with significant increase in sampling rates, reduction in cost, and performance improvements of interrogation equipment make FBG sensors a breakthrough technology for mitigation of LV vibrations, leading to improved vehicle performance and capability.

Vibration suppression control has been applied in civil engineering, flexible robot arms and space satellite attitude control in past decades. In [15], Lin presented piezoelectric damp-modal actuators/sensors to control a cantilever laminated composite beam with integrated sensors and actuators. By varying the voltage supplied to the actuators, modal control forces are applied to increase damping ratio, effectively attenuating vibrations in the laminate beam.

Balas [16] presents feedback control with multiple point forces and sensors. The method controls a large dimensional system with a much smaller dimensional controller since computer limitations and mode error could restrict control to a few critical modes. The possible effect of ignoring residual modes is discussed, and augmenting the controller with a pre-filter is proposed to eliminate the most deleterious residual mode effects.

H-infinity loop shaping [17,18,19,20] has been extensively used for vibration suppression in flexible spacecraft. Elgersma et al. [17] designed controllers for space station attitude control and momentum management. Performance tradeoffs between stabilization, attitude regulation and momentum magnitudes were explored using H-infinity optimization. The resulting controllers stabilize the unstable gravity gradient torques, keep altitudes and momentums due to small aero disturbance torques, and are robust to uncertainties in moments of inertia of the space station. Hyde and Glover [18] developed a flight controller for the longitudinal motion of a Harrier jump jet using H-infinity techniques. Gadewadikar et al. [19] presented a helicopter 3D attitude controller with guaranteed performance based on shaping the attitude step response using H-infinity output-feedback to achieve tracking in all three attitude states. Chen et al. [20] propose a controller that takes both magnitude and rate constraints of the actuator dynamics into account. The effectiveness is verified through the achievable H-infinity performance and time response simulations with specified constraints on magnitude and control torque rate.

Most results of previous research on flexible spacecraft are based on simulations. Using a flexible cantilever beam as surrogate enables experimental assessment of the control system [21,22,23,24,25]. Lenz et al. [21] showed a distributed parameter H-infinity controller for a flexible Euler–Bernoulli beam. Sano [22] demonstrated a finite-dimensional H-infinity controller with measurement feedback using flexible beam equations. Petersen and Pota [23] presented a robust controller based on minimax LQG control, illustrated by robust vibration control of a flexible cantilever beam subject to white noise disturbances that minimizes total vibration energy; the control is robust against uncertainties introduced by neglecting high-order beam modes. Scholte and D’Andrea [24] applied distributed control to the noise radiation of a flexible structure: penalizing velocity states results in a reduction in noise radiation. The distributed controller outperforms a local velocity feedback controller, and stability is guaranteed even when the actuators are not point actuators. Kircali et al. [25] developed active vibration control of a cantilevered smart beam by using the assumed-modes method. A spatial H-infinity controller was designed for suppressing the first two flexural modes of the beam, demonstrated both by simulations and experimental implementation. 

Optical frequency domain reflectometry (OFDR [26,27]) can achieve spatial resolution in the order of tens of microns and can work with fiber Bragg grating sensors. OFDR is truly distributed and has the ability to interrogate any point within continuous media, as opposed to FBG arrays that can only provide measurements on a finite number of points. On the other hand, OFDR requires the use of tunable laser sources with long coherence length, high degree of linearity in optical frequency sweeping, and wide spectral range, possibly resulting in highly expensive systems that may not operate reliably in an aggressive environment such as a launch vehicle application.

This paper builds on several of the concepts previously mentioned to propose a method for LV vibration suppression based on H-infinity loop shaping techniques and the use of fiber Bragg grating sensor arrays for the real time estimation of natural frequencies and mode shapes [28]. If the bending vibration of a slender rocket can be controlled, mass reduction and improved flight performance can be achieved. FBG sensors can also be used as sensor arrays for control of independent modes.

The experimental model presented corresponds to a cantilever beam supported on a moving base that can apply horizontal force to the beam as forced displacement. In a real launch vehicle application, horizontal forces acting at the base of the vehicle would be provided by horizontal components of the thrust engine forces, as in thrust vector control. Future launch vehicles may incorporate side thrusters in other locations, which is represented by the cold gas thrusters used on the paper. The similarities in actuation between the prototype and a launch vehicle suggest the applicability of the proposed method in an actual LV application. External forces can be considered as unknown disturbances; the control actuators can either be thrust vector, cold gas or small solid propellant thrusters.

## 2. Materials and Methods

### 2.1. Real-Time Control of a Flexible Beam

One of the objectives of this study is the design of a scaled test article representative of the structural dynamics of a slender launch vehicle, by ensuring the first three modal frequencies fall within ranges of those of an actual launch vehicle. The target frequency range is based on the natural frequencies of an ideal cantilever beam [29], Equation (1):
(1)$$\omega_{n}={\left(\beta_{n}l\right)}^{2}\sqrt{\frac{EI}{\rho Al^{4}}}$$
where *β_n_* is a constant; (*β_n_*)^2^ = 1.875, 4.694, and 7.855 for the first three natural frequencies of an ideal cantilever beam of Young’s modulus *E*, moment of inertia *I*, mass density *ρ*, cross-sectional area *A*, and beam length *l*. From (1), the relationship between the 1st, 2nd and 3rd modal frequencies is determined. A frequency range for the first modal frequency is chosen that brings the second mode within range of actual launch vehicles, with dimensions feasible to implement on a tabletop cantilever beam experiment. The fundamental vibration modes in launch vehicles are vibrations in the pitch plane. For this reason, a rectangular cantilever beam was chosen as the test specimen. An example of a next generation slender launch vehicle is shown on Figure 1a, the instrumented test article, designed to have similar structural dynamics characteristics, is shown in Figure 1b,c.

The test beam includes two pairs of pressurized air side thrusters, used either to control or excite vibrations in the beam. Pressurized air is supplied by a manifold shown in Figure 1. The test article is mounted on an air bearing driven by linear electromagnetic actuators, which is also used to either disturb or control the test article. The FBG strain sensors used in the test article are the Micron Optics os3200 non-metallic optical strain gages. An array of 18 FBG sensors were connected to the Micron Optics sm130 optical sensing interrogator; more details on the os3200 sensors can be found in [30].

### 2.2. Mathematical Model of the Flexible Beam

The test article is a slender cantilever beam with a rectangular cross-section, whose first three natural frequencies are similar to those of a LV. To make bending the predominant mode of deflection, the beam’s length (*l*), needs to be much longer than the width (*w*), and the width needs to be much wider than the thickness (*h*). The area moment of inertia of the beam’s cross section determines the main axis of bending, the 1^st^ natural frequency (*ω*_1_) and 1^st^ bending mode. As thickness decreases, the beam deflects more easily in the direction perpendicular to w compared to the orthogonal direction h:
(2)$$\omega_{1}=\sqrt{\frac{k}{m}},\;k=\frac{3EI^{3}}{l^{3}}=\frac{Ewh^{3}}{4l^{3}}$$
where *m*, *k*, *w* and *h* are the beam’s mass, stiffness, width and thickness, respectively, and *ω_n_* its *n*th natural frequency in the predominant bending mode. Increasing the length (*l*) while keeping the area moment of inertia constant decreases the natural frequencies.

To simulate the closed loop system, a model of the flexible beam is required. Finite element analysis software (ANSYS 15.0) was used to create a model of the test article that can be implemented in, MATLAB/Simulink (R1014.b) to develop control solutions. The solution of the FEA provides the matrices of the system of equations of motion:
(3)$$mẍ+cẋ+kx=0,\;J\overset{˙˙}{\theta }+c\dot{\theta }+k\theta =0$$
where *c* and *J* are the beam’s viscous damping matrix and rotational inertia matrix, and *x*, *θ* are translational and rotational vectors of degrees of freedom for each beam element. The beam element used is a line element based on Timoshenko’s beam theory: for thick beams (a beam whose cross-sectional area is not small compared to its length), the effects of shear deformation and rotational inertia must be considered [29,31]. The beam is considered a cantilever with one end fixed in all 6 degrees of freedom (DOFs) and the other end free in all 6 DOFs. The model does not incorporate thermal or fluid effects: room temperature is assumed constant and drag effects from beam motion are negligible.

### 2.3. Finite Element Model of the Flexible Beam

The finite element model reflects the entire test article, including all components mounted on the beam: thrusters, pressurized air plumbing, and wiring harness. The FBG sensor array has negligible mass compared to the other components. The mechanical coupling caused by the air supply lines was also neglected since tubing stiffness is negligible compared to beam stiffness. To model the wiring harness, the linear mass density of the harness was calculated and modeled as additional point masses placed on the beam every 2 elements: the wiring harness mass locations does not overlap with valve and nozzle mass element locations, while maintaining an even distribution along the beam. Since the distribution of wire is not uniform over the length of the beam, the point mass values were adjusted accordingly.

Since the test article is long and slender, the stress introduced by gravity when the beam is oriented vertically cannot be ignored. To incorporate this effect, the FEA model is implemented using a pre-stressed modal analysis with large deflection, where the stress induced by the gravity load on the beam generates an effective stiffness that adds to the inherent material stiffness.

The cold gas thrusters attached to the beam have substantial mass, located at an offset distance from the beam’s neutral axis. Their inertia therefore needs to be included. The inertia of the thrusters was determined experimentally and was calculated based on the principle of equivalent rotational mass: for a body that undergoes both rotation and translation the total inertia may be represented as the sum of both contributions [29].

The final model reflects the fully integrated test article, including thrusters, FBG sensors, plumbing and wiring harness. The frequencies for the resulting beam model are shown in Table 1. The first five modal frequencies of the FEA model can be compared to experimental measurements obtained using the FBG sensor array mounted on the beam and a capacitive sensor (Figure 2).

Table 1 shows agreement between experimental resonant modes of the test setup and the resonant modes estimated from the FEA model by two different methods within less than 2% differences. The FEA model will be used for controller synthesis and simulations. The main source of model error comes from neglecting the stiffness from the plumbing/tube interaction between thrusters and supply manifold. In even numbered modes, the node locations are such that the restoring forces generated by the 2 pairs of tubes are 180° out of phase, and cancel out. In odd-numbered modes, node location and restoration forces cause that the two forces are in phase, adding an unbalanced force opposing motion. The open loop vibration spectrum of the test article, measured by capacitance sensor and FBG array, is shown in Figure 2.

To illustrate that the FBG sensor system can capture the dynamics of the flexible beam, its spectral estimation ability is compared to that of a high bandwidth capacitive sensor. The spectral comparison of the FBG sensor and capacitive sensor measurements, when the beam is subject to noise base excitation, is shown in Figure 2: modal frequencies found using the FBG sensors are identical to those found using the capacitive sensor.

### 2.4. Fiber Bragg Grating Sensor Arrays

The fiber optic sensor (FOS) system measures strain along the beam using fiber optic Bragg gratings. FOS enables measuring a large number of strain sensing points without the delay normally associated with multiplexing a large number of analog strain gauges. The weight of optic fibers is almost negligible compared to the weight of wiring harnesses for conventional strain gauges, making the incorporation of a large number of strain sensors into a launch vehicle possible. Fiber optic strain sensors are multiplexed at the speed of light, minimizing measurement delay associated with polling multiple sensors.

A fiber Bragg grating sensor consists of localized periodic changes of the refractive index at the core of an optical fiber, generated by exposure to an intense UV interference pattern. This periodic change, called grating, acts like a band stop filter: the grating reflects a specific wavelength component of the injected light back, while allowing the rest of the light to travel through the grating unobstructed. The wavelength reflected back (Bragg wavelength) relates the wavelength reflected to the grating period by the Bragg condition:
(4)$$\lambda_{B}=2n_{e}\Lambda$$
where *λ_B_* is the Bragg wavelength of the FBG sensor, *n_e_* is the effective refractive index of the fiber core, and *Λ* the grating period. The wavelength given by the Bragg condition is reflected by the grating, while other wavelengths pass through with minimal attenuation, as shown in Figure 3. Light continues through the fiber until it hits the next grating, with a different Bragg wavelength. Changes in a physical property at the sensor location (either strain or temperature) result in a change in the reflected wavelength. By measuring this wavelength change relative to the nominal wavelength of the FBG sensor within a fiber, the change in either strain or temperature can be measured.

Fiber Bragg grating sensors operate by detecting changes in the fiber’s reflective index at the sensor’s location. The interrogator works by sending a broadband light pulse into the fiber array and measuring the reflected wavelength peaks at the given sensor locations. The shift in wavelength corresponds to a change in physical properties at the sensor location. Changes in Bragg wavelength due to strain and temperature are shown below as *ε*; the strain corresponding to the measured wavelength shift is Δ*λ*; the strain measured due to temperature changes is denoted as *ε_TO_*:(5)$$\varepsilon =\left(\frac{\Delta \lambda }{\lambda_{0}}\right)\frac{1\times 10^{6}}{F_{G}}-\varepsilon_{TO},\;\varepsilon_{TO}=\Delta T\left[\frac{C_{1}}{F_{G}}+CTE_{s}-C_{2}\right]$$
where *λ*_0_, *C*_1_, *F_G_*, *CTE_S_*, and *C*_2_ are constants provided by the manufacturer. In addition to a fiber optic strain sensor array, two fiber optic sensors are included in the test setup to enable temperature compensation. The temperature change between these sensors (Δ*T* in Equation (5)) is used to compensate strain measurements for temperature changes.

The FBG sensor arrays are connected to the Micron Optics sm130 FBG interrogator (Atlanta, GA, USA), Figure 3b. The interrogator provides up to four optical channels, with up to 80 sensing points per channel. The sensors are all sampled simultaneously, with a sampling rate of 1000 Hz. This high sampling rate, combined with the fact that the sm130 interrogator provides a non-buffered output, makes it very suitable for real-time feedback control, where rapid sampling with minimal sensor delay is critical for good closed loop performance. The interrogator works by sending broadband light pulses into the fiber arrays and measuring the reflected wavelength peaks. The wavelengths of these peaks are communicated to a host PC using the transmission control protocol (TCP/IP).

### 2.5. Flexible Beam Displacement Model

A modal approach is used to obtain a displacement–strain relationship: using displacement mode shapes and strain mode shapes from the FEA model, a displacement–strain transformation matrix can be calculated. By assuming that displacement and strain of structures can be expressed using a finite number of mode shapes, *N*, the displacement and strain can be expressed as
(6)$$\left\{d\right\}=\left[\Phi_{N}\right]\left\{\eta_{N}\right\},\;\left\{s\right\}=\left[\Psi_{N}\right]\left\{\eta_{N}\right\}$$
where {*d*}, [*Φ_N_*], {*s*}, [*Ψ_N_*] and *η_N_* represent the vector of structural displacements, the matrix of displacement mode shapes, the strain vector, the matrix of strain mode shapes and the vector of modal coordinates, respectively. From Equation (6), the transformation between strain and displacement is obtained, yielding:(7)$$d_{n\times 1}={\left[T_{ds}\right]}_{n\times M}s_{M\times 1}={\left[\Phi \right]}_{n\times N}{\left({\left[\Psi \right]}_{N\times M}^{T}{\left[\Psi \right]}_{M\times N}\right)}^{-1}{\left[\Psi \right]}_{N\times M}^{T}s_{M\times 1}$$
where *n* is the number of points in the beam whose displacement is estimated, *N* the number of deflection mode shapes considered, and *M* the number of strain mode shapes considered. The matrices of displacement mode shapes and strain mode shape are obtained from the FEM modal analysis solution. This method of predicting displacement using strain measurements was demonstrated by Jiang et al. [28].

The strain mode shapes can only be determined off-line for the baseline LV structure. Changes in mass or stiffness that lead to changes in the strain mode shapes become model uncertainty, since the strain mode shapes cannot be recalculated in real-time. The baseline strain mode shapes must therefore be regarded as an uncertain model with bounded uncertainty. Model deviations during real-time operation need to be compensated by the robustness of the controller, which highlights the importance of using a robust controller design approach such as H-infinity.

### 2.6. Thruster Model

Two pairs of computer-controlled cold gas thrusters that can apply a specified force-time history (impulse) to the flexible beam are used to either reduce vibrations in the flexible beam, or apply external force excitation. The location and orientation of the thrusters is shown in Figure 1. Each thruster consists of three parts: a nozzle manufactured by fused deposition printing, a section of tube, and an on–off electro valve. The valve switches the compressed air intake on or off, and the nozzle directs the flow of compressed air perpendicular to the beam. Nozzle diameter is selected to choke the flow under nominal steady-state operation, which provides a known mass flow and hence a known thrust force. The top pair of thrusters mounted on the beam is shown in Figure 4.

A model of the thruster system is required. During the transient start-up of nozzle flow, there is a delay until the equilibrium pressure within the tube is reached, and therefore a delay until the nozzle delivers a steady-state mass flow (constant thrust). To control the beam bending dynamics, the model must take into account the transient start-up associated with the valve turning on, as well as the transient decrease in thrust when the valve is turned off. A mathematical model of the transient thrust in time was developed based on valve dynamics, as well as the filling and venting of air from the supply tube and the volume behind the nozzle exit, which together act as a reservoir.

The model considers filling of the reservoir from the supply pressure when the valve is turned on and the venting of air to produce thrust through the nozzle orifice. At each time step, air flows into the reservoir causing the reservoir pressure to increase. Simultaneously, the reservoir is allowed to vent through the nozzle orifice, which decreases the pressure of the reservoir. The governing equations for compressible flow of a gas through an orifice is shown in Equation (8), along with the change in reservoir pressure *P_i_*:(8)$$ṁ=\frac{MA\left(t\right)P\sqrt{\gamma }}{\sqrt{RT}{\left(1+\frac{\gamma -1}{2}M_{valve}^{2}\right)}^{\frac{\gamma +1}{2\gamma -2}}},\;\;P_{i-1}\pm \left(\frac{ṁRT}{V}\right)\Delta t=P_{i}$$
where *ṁ* is the mass flow rate, *P* the reservoir/supply total pressure, *R* the specific gas constant for air, *T* the gas temperature, Δ*t* the time step, *V* the reservoir volume of the length of tube plus the volume behind the orifice within the nozzle, and *γ* the specific heat ratio of air, respectively. Equation (8) shows the fluid model is dependent on pressure and Mach number. The equation for mass flow is only applied at the boundaries of the reservoir, and the equation for pressure is only applied in the reservoir. The model makes the following assumptions: Frictional pipe losses are incorporated as an offset to tune the model, air supply pressure remains constant, choking only occurs at the inlet and exit of the reservoir, and the reservoir has a constant cross section. Valve dynamics are modeled as a first-order differential equation, since valves are solenoid types:(9)$$A(t)=A_{max\;}e^{-\tau t}$$
where the opening time is defined as a time constant, τ, related to the inductance of the solenoid. The variable controlled in time is the valve’s flow area *A*(*t*), *Amax* denotes the valve flow area at full open as time t approaches infinity.

The valve model was validated using experimental measurements of thrust, as shown in Figure 5. Thrust measurements are performed by mounting the nozzle on a multi-axis force sensor. Figure 5b shows the measured current through the valve coil (blue), the measured thrust (green), and the thrust predicted by the model (red). To match the measured thrust with the model, adjustments are made to the nominal values of the open/close time and orifice size of the valve, provided by the valve manufacturer (Festo).

Figure 5 shows the response of both the actual valve and the mathematical valve model to coil current input. The total impulse of the measured thrust is 0.0643 N, and the total impulse of the thrust predicted by the model is 0.0632 N. The difference between predicted impulse and measured impulse is 1.67%. The FESTO MHJ9-MF valves used in the experimental setup are on-off valves: the valves can only switch between fully open and fully closed. To generate any level of thrust between zero and the maximum thrust, the valves are pulse width modulated (PWM). The thruster model includes a transport delay, and both valve and flow dynamics introduce additional delays in its operation. Since the thrust model is relatively accurate (Figure 5b) the effect of the thruster dynamics can be considered in controller design and therefore compensated by the controller.

### 2.7. ‖H‖∞ Control Algorithm

The background of robust control is the small gain theorem [32]. The small gain theorem can be explained by considering a loop with transfer function *H*. This loop is stable if *H* is stable, which is the case if *‖H‖∞* is smaller than one. The *‖∙‖∞* norm denotes the largest singular value, which is equivalent to the largest gain of the system. When *‖H‖∞* is smaller than one, the Nyquist diagram of the transfer function H can never encompass the −1 point. The Nyquist criteria states that if an open-loop curve stays within the unit disc, the system will always be closed-loop stable, since all the closed-loop poles will be in the left half plane. Having a gain smaller than one is a sufficient but not a necessary condition for a system to be stable.

One of the motivations for choosing H-infinity control synthesis over classic control methods is the fact that H-infinity controllers are able to manage single-input multi-output (SIMO) plants. Using a single sensor to calculate the desired control action has a significant influence on the achievable performance of a controller. In the cantilever beam test article, multiple sensor measurements can be used to calculate the required control action, which has significant advantages when controlling multiple modes. To demonstrate the effect of using multiple sensors, the controller objective is set to suppressing the first two modes, using displacement measurements co-located with the actuator locations. To obtain robust stability and shape the sensitivity, an augmented plant for the mixed sensitivity problem is defined, as shown in Figure 6.

The combined filters of the augmented plant determine the controller input weight, and are defined in a way that the product of the actuator saturation (*V_d_*) and the uncertainty weight (*W_u_*) cover the unmodeled plant dynamics resulting from the model order reduction, as shown in Figure 7.

The output error weight (*W_e_*), which determines the characteristic of disturbance rejection, is specified as the inverse of the allowed magnitude of a specific output error at each frequency. Since this controller is designed to suppress both 1st and 2nd modes, *W_e_* contains two inverse notches corresponding to the first two eigenfrequencies of the beam system, and a lowpass filter to suppress steady state error. The filter weights used for the H-infinity controller synthesis are shown in Figure 8.

Using this method, two MISO H-infinity controllers are designed, which calculate the control action for both thrusters. The first calculates the control action for the top thruster based on the SIMO plant with displacement output at the top and middle thruster location, and a force input at the top of the beam. The second MISO H-infinity controller calculates the control action for the middle thruster, based on the SIMO plant with displacement output at the location of the top and middle thrusters and a force input at the location of the middle nozzle.

Both controllers (individually) are able to reduce vibration in both modes, since they use measurement points at both nozzle locations. Using both controllers in a double-MISO configuration results in better performance and reduces vibration at both modal frequencies. The goal of the control system is to provide 90% vibration attenuation compared to open-loop behavior.

## 3. Results

The double-MISO controller was first simulated using Simulink. The simulation includes the double-MISO controller, the pneumatics model for both top and middle thruster, the state space beam model with force inputs for top and middle thruster, the base motion excitation input and the model for the base actuator (linear motor)

The simulation results obtained from Simulink are validated using the experimental beam setup. Figure 9 and Figure 10 show the system response to a disturbance in the form of an initial deflection in the first mode and second mode, respectively. The plots show the open-loop response of the beam system (blue) and the double-MISO H-infinity-controlled closed loop (red). Simulation results are shown on the left, and experimental validation under same initial conditions are shown on the right. The top figures show displacement at the top thruster pair (at the tip of the beam, 1.57 m); the bottom figures show the displacement at the bottom thruster pair, mounted 0.91 m from the base of the beam.

The response of the closed-loop system after initial deflection, shown in Figure 9 and Figure 10, shows rapid settling time in both first and second mode, respectively. The settling time of the closed-loop system when the beam is initially displaced in the first mode shape is 2.2 s, with an initial displacement in the second mode shape of 1.8 s.

Figure 11 and Figure 12 show results when the beam is continuously excited with sinusoidal base motion using the base actuator, at the first modal frequency and the second modal frequency, respectively.

The closed-loop results (red) are compared against the open loop results (blue). Simulation results are shown on the left column, the experimental validation of these results is shown on the right column. The top figures show displacement at the top thruster pair (at the tip of the beam, 1.57 m from the base), bottom figures show the displacement at the bottom thruster pair, mounted 0.91 m from the base of the beam.

The proposed controller was also tested using a step-base motion excitation, with results shown in Figure 13. A step excitation was chosen since it excites all frequencies in the beam: the controller needs to suppress vibrations at all frequencies at the same time. The open-loop response of the beam (blue) is compared to the controlled closed loop response (red). Simulation results are shown on the left; experimental results under the same conditions are shown on the right. The top figures show displacement at the top thruster pair (at the tip of the beam, 1.57 m from the base); the bottom figures show the displacement at the bottom thruster pair, mounted 0.91 m from the base of the beam.

The frequency response of the proposed controller is shown in Figure 14, showing notches at the first two modal frequencies. The frequency response of the system is shown in Figure 15, including both the full-order SIMO plant and reduced order SIMO plant.

## 4. Discussion

Experimental data match simulation results closely, with an RMS error smaller than 5 mm. In continuous base excitation at first modal frequency, (Figure 11) the oscillation of the closed-loop system is smaller in the experiments (right) than in the simulations (left), which increases the RMS error, even though this result is desirable.

A small oscillation remains after settling, most noticeably in the second mode experiments (Figure 10). The controller is able to damp out the free vibration, though after this initial damping, a small amplitude oscillation with a frequency equal to the first resonant frequency remains. This oscillation damps out at the same rate as the open-loop vibration. This is related to the Bode diagram of the controller (Figure 14), showing notches at the natural frequencies; controller action at this frequency is small.

When the beam is continuously excited, energy is constantly added to the system using the linear base actuator. Figure 11 shows beam vibration when the first modal frequency is excited continuously. Dual-MISO control reduces peak to peak vibration by 94%, compared to the open loop response of the system excited with the same base motion.

The hardest case to control is when the second modal frequency is continuously excited using the base actuator, as shown in Figure 12. The simulation shows a vibration reduction of 90% compared to the open-loop response of the system excited with the same base motion, which meets the vibration reduction requirement. The best experimental vibration reduction achieved with the double MISO controller when the second resonant mode of the beam was exited was 80%.

The issue faced with continuous excitation at the second resonant frequency is that control performance is physically limited by the power available from the thrusters at this frequency. The vibration in the beam cannot be entirely controlled because the thrusters are incapable of matching the strain energy added by the base at that frequency. This can be seen in Figure 12 by the slight increase in oscillation amplitude of the closed-loop-controlled system. A solution would be to decrease the amplitude of the base excitation.

Beam displacement after a step input is applied to the experimental setup using the base actuator (Figure 13) shows the same rapid damping performance of the control system as when the beam is subject to initial deflection. The controller is able to damp out vibrations at all excited frequencies with a setting time of 0.6 s.

## 5. Conclusions

This paper presents a method to reduce vibrations in slender launch vehicles using fiber Bragg grating strain sensor arrays and a robust controller. A test article with natural frequencies representative of a launch vehicle was designed and built: the dominant resonant frequencies were chosen to fall within boundaries of a nominal launch vehicle.

FBG sensor arrays are a breakthrough technology for measuring strain in real time in flexible structures. FBG sensor arrays provide the capability of measuring strain at a large number of sensing points multiplexed at the speed of light, which virtually eliminates the measurement delay normally associated with polling multiple sensors. FBG arrays are also much lighter than traditional wiring harnesses for strain gauges and are insensitive to electromagnetic interference, and the fiber diameter allows installation almost anywhere on the structure. The only downside of FBG strain sensors over traditional strain gauges is their sensitivity to temperature fluctuations, but this can be addressed using FBG temperature sensors for compensation of strain measurements.

Finite element modeling was used to model and simulate the vibrating test article. The error between the FEM model and experimentally determined resonant frequencies is reduced to less than 2% by including stress stiffening and inertia into the model.

The displacement array in the flexible beam setup can be calculated using displacement and strain modal shapes obtained from beam’s FEM model plus the vector of FBG strain sensor measurements along the beam. Displacement information is used in the feedback control system to attenuate vibrations in the experimental setup.

A robust controller based on the FEM model was developed to minimize vibration in the test article. The controller uses the vector of strain measurements from the fiber Bragg grating sensor array, converted to displacement, to calculate the required control action at the thrusters. With this controller, the continuously excited first mode under base excitation shows a 94% reduction in peak–peak vibration.

The continuously excited second mode under base excitation shows an 80% reduction in peak–peak vibration in the test article, while simulation of the same case shows a 90% reduction in peak–peak vibration. The control performance of the second mode is physically limited by the power available from the thrusters at that frequency.

It is envisioned that future structural control methods can be based on FBG sensor arrays. Candidate methods include linear control techniques (as benchmark), sliding mode control, and L1 adaptive control. The developed test setup enables future tests of vibration control using base motion as the actuator while using the side thrusters to excite vibrational modes in the beam. This approach would simulate thrust vector control (TVC), using the gimbal engine of a launch vehicle to control vibrations induced in the rocket body by cross winds or aerodynamic forces.

## Figures and Tables

**Figure 1 sensors-25-00204-f001:**
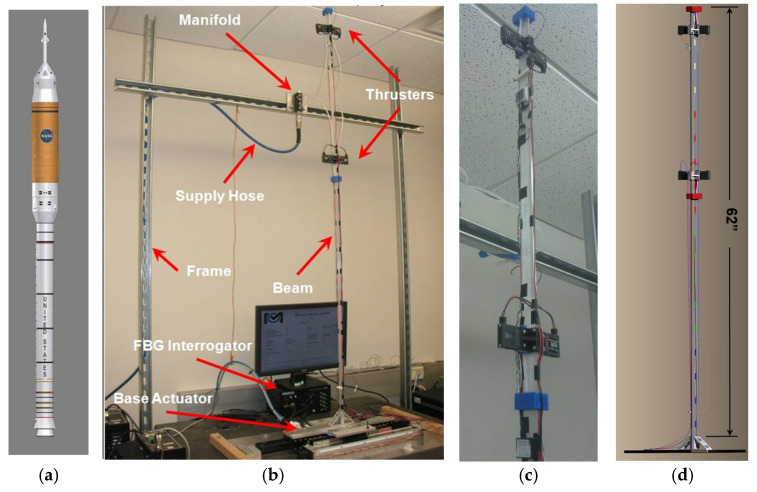
Instrumented flexible beam test article representative of slender launch vehicle structural dynamics. (**a**) slender launch vehicle example, (**b**) experimental setup for active control of flexible structure with similar natural frequencies to (**a**), (**c**) cold gas thrusters shown on test specimen, (**d**) Placement of FBG sensors on test specimen.

**Figure 2 sensors-25-00204-f002:**
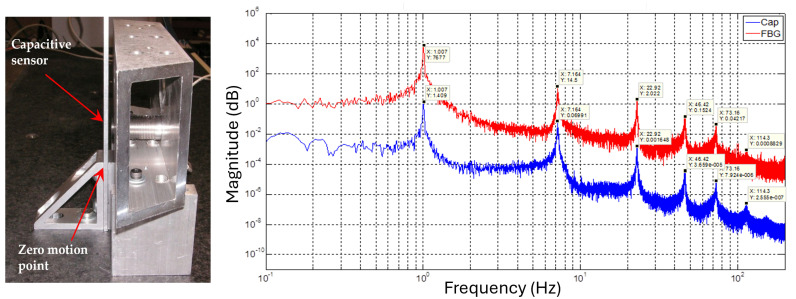
Vibration spectrum of the test article measured with capacitance probe and FBG.

**Figure 3 sensors-25-00204-f003:**
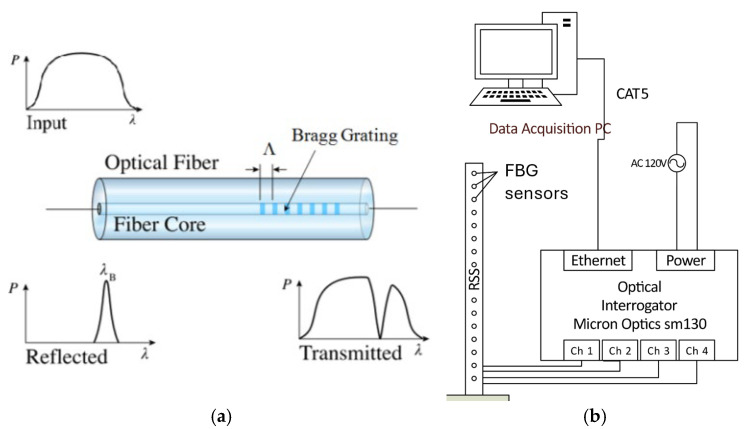
(**a**) Operating principle of the fiber Bragg grating (FBG) sensor. (**b**) FBG sensor interrogation system. In this study, only one fiber channel was used.

**Figure 4 sensors-25-00204-f004:**
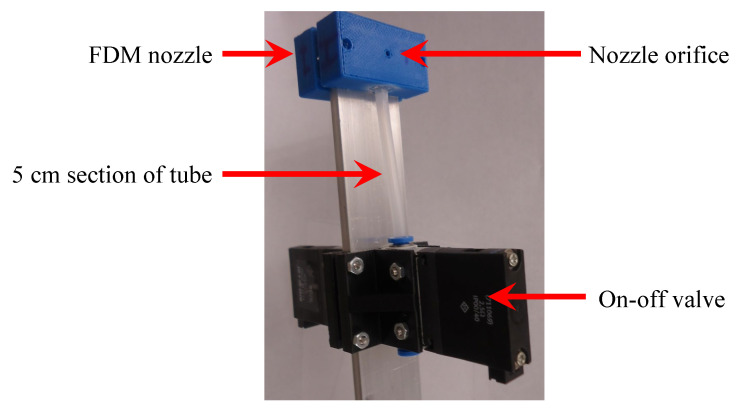
Thruster assembly and components.

**Figure 5 sensors-25-00204-f005:**
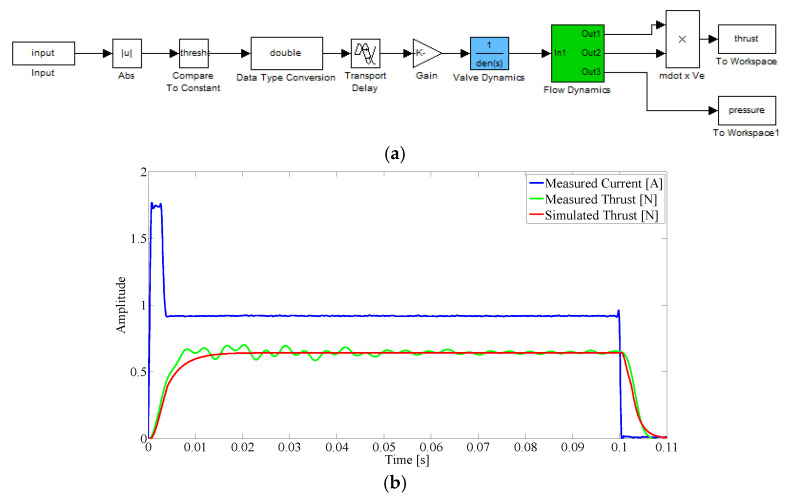
(**a**) Thruster transient model, (**b**) thruster model response vs. measured thrust for experimentally measured current input.

**Figure 6 sensors-25-00204-f006:**
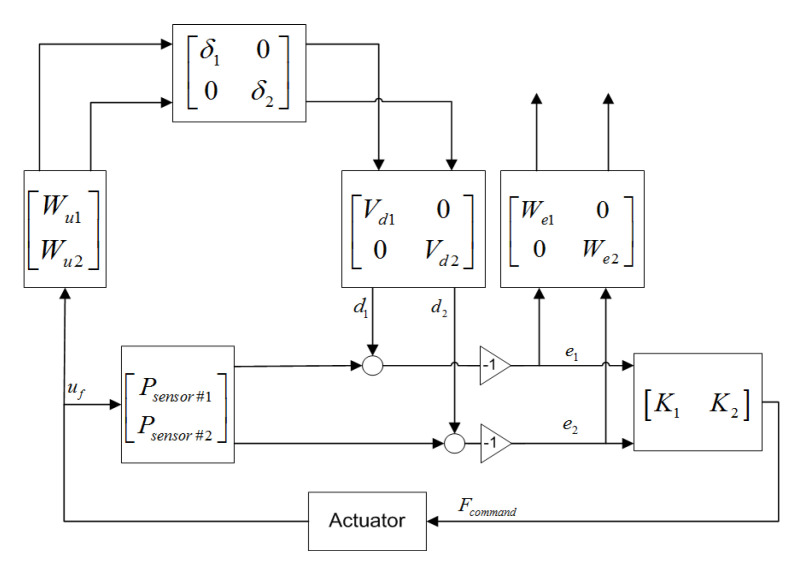
Augmented SIMO plant for mixed sensitivity problem.

**Figure 7 sensors-25-00204-f007:**
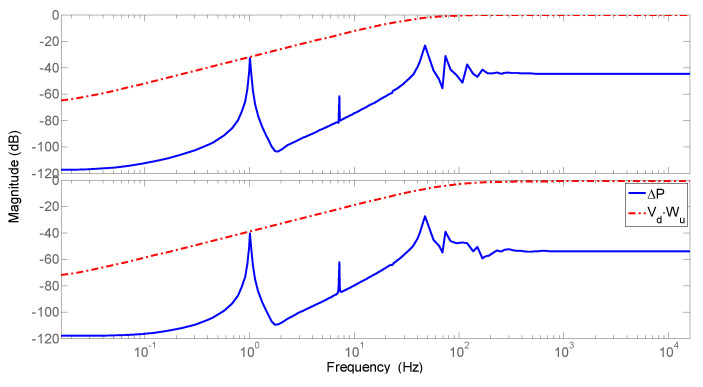
Definition of MISO controller input weights.

**Figure 8 sensors-25-00204-f008:**
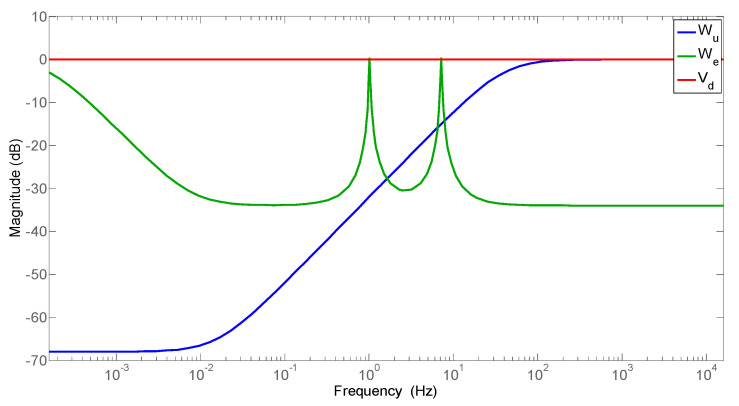
Weighting filters used for H-infinity synthesis.

**Figure 9 sensors-25-00204-f009:**
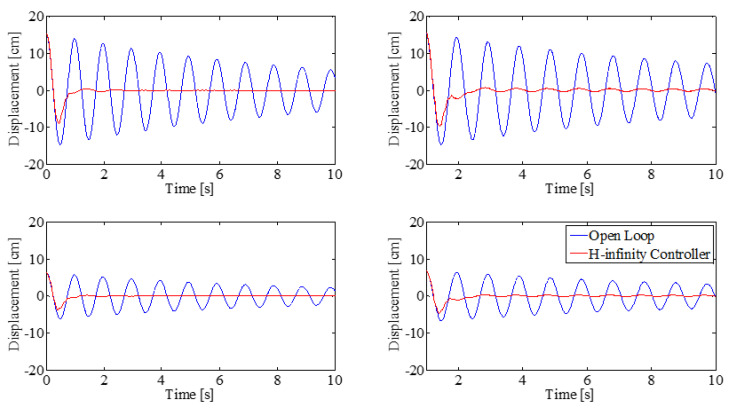
Double-MISO H-infinity controller response with initial deflection in first mode shape, simulation results (**left**), experimental results (**right**). Top row: displacement at top thruster location, Bottom row: displacement at bottom thruster location.

**Figure 10 sensors-25-00204-f010:**
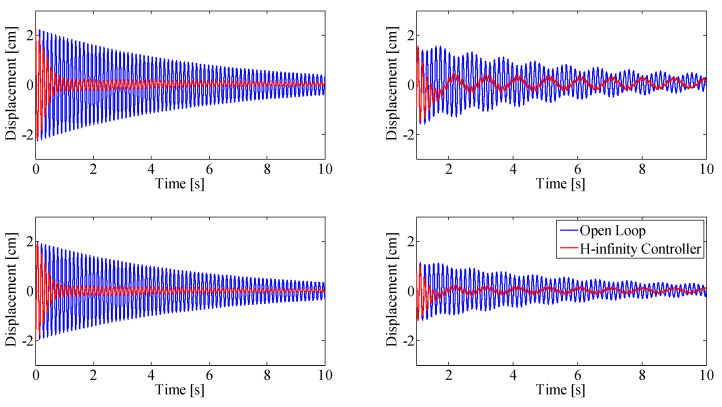
Double-MISO H-infinity controller with initial deflection in second mode shape, simulation results (**left**), experimental results (**right**). Top row: displacement at top thruster location, Bottom row: displacement at bottom thruster location.

**Figure 11 sensors-25-00204-f011:**
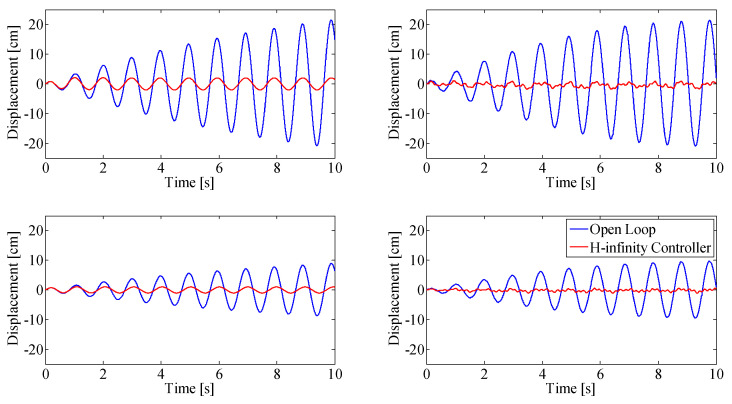
Double-MISO H-infinity controller with continuous base excitation at first modal frequency. Simulation results (**left**), experimental results (**right**). Top row: displacement at top thruster location, Bottom row: displacement at bottom thruster location.

**Figure 12 sensors-25-00204-f012:**
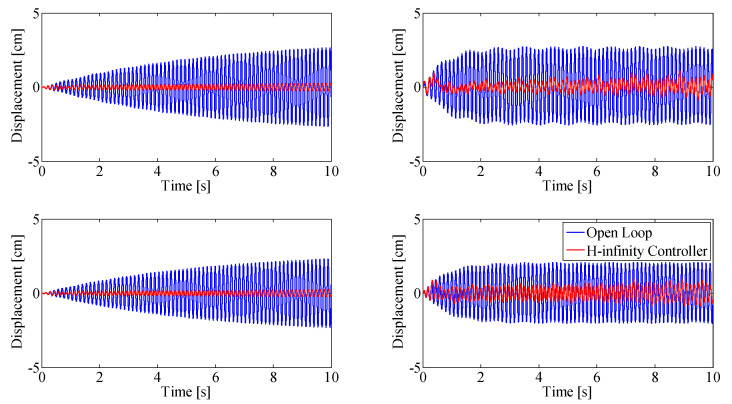
Double-MISO H-infinity controller with continuous base excitation at second modal frequency. Simulation results (**left**), experimental results (**right**). Top row: displacement at top thruster location, Bottom row: displacement at bottom thruster location.

**Figure 13 sensors-25-00204-f013:**
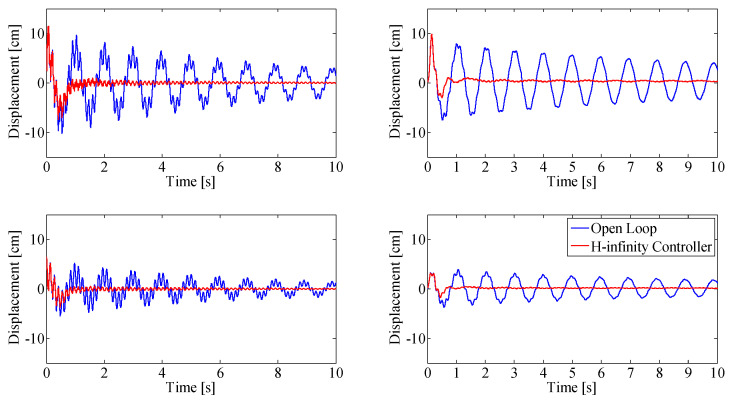
Performance assessment of the double MISO H-infinity controller after step base motion excitation. Simulation results (**left**), experimental results (**right**). Top row: displacement at top thruster location, bottom row: displacement at bottom thruster location.

**Figure 14 sensors-25-00204-f014:**
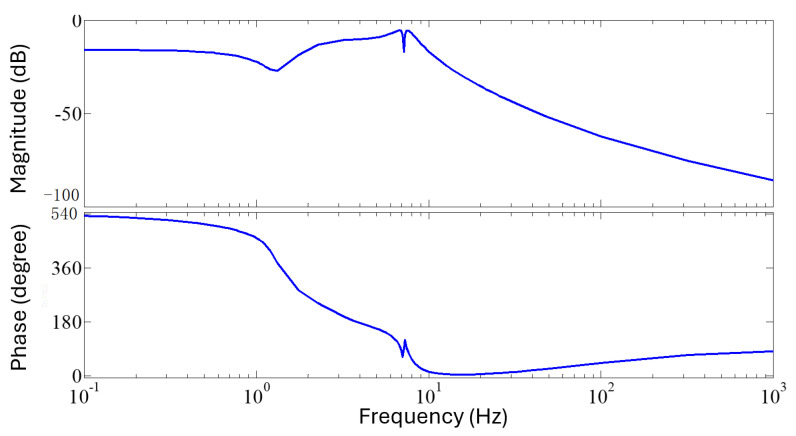
Bode diagram of the controller, showing notches at the natural frequencies.

**Figure 15 sensors-25-00204-f015:**
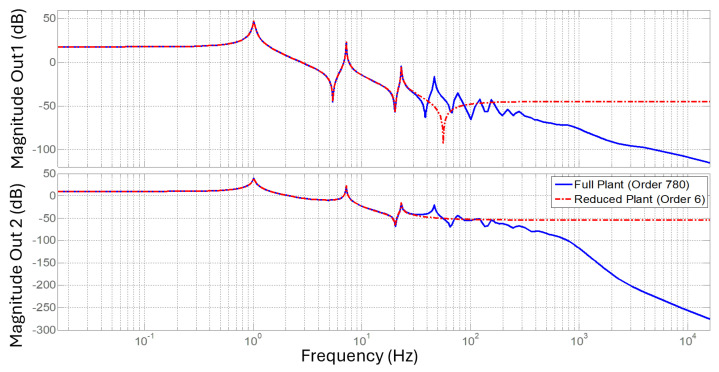
Frequency response of the full order SIMO plant vs. reduced order SIMO plant.

**Table 1 sensors-25-00204-t001:** Comparison of modal frequencies: FEA model vs. experimental measurements.

Mode	Modal Frequency (Hz)	FEA Model, Block Lanczos Method	% Diff	FEA Model, QR Damped Method	% Diff
1	1.03	1.0146	−1.49%	1.0145	−1.50%
2	7.19	7.1902	−0.06%	7.1902	−0.06%
3	22.89	23.142	1.07%	23.141	1.07%
4	46.18	46.401	0.47%	46.395	0.46%
5	72.99	74.355	1.85%	74.331	1.82%

## Data Availability

The data sets presented in this article are not readily available, but results can be replicated under comparable experimental conditions. Requests to discuss replication of the results presented herein may be directed to the corresponding author.

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
