# Peer review of "Vibration Control of Flexible Launch Vehicles Using Fiber Bragg Grating Sensor Arrays"

_sensors, 2025, doi:10.3390/s25010204_

Round 1

Reviewer 1 Report

Comments and Suggestions for Authors

The authors have provided an interesting and timely work with an important practical component. However, I would like to ask the authors to pay attention to some of my comments. I consider it appropriate to publish this article after the shortcomings have been corrected. My comments on this article are given below:

1. The abstract to this work states that distributed measurements will be presented. I would not agree with this term, because in fiber optic terminology, distributed measurements are usually understood as measurements using optical reflectometry. Measurements performed using fiber Bragg gratings are usually called quasi-distributed or pointwise measurements.
2. The introductory part to this article seems inconsistent. Thus, usually, when mentioning some scientific work, it is customary to first talk about the results achieved by the scientific group, and then explain why these results are not suitable for solving your problem. Such logic will allow you to formulate the motivation for the work and outline its scope by the end of the introductory part.
3. In addition, in the introductory part it would be appropriate to compare the approach chosen by the authors with the already mentioned distributed systems, for example, based on the principles of optical reflectometry in the frequency domain. When using fibers without special reflectors, this method allows achieving spatial accuracy of several tens of microns, while the method also allows working with fiber Bragg gratings. Perhaps you can find suitable articles in this review: [http://dx.doi.org/10.3390/s24165432]. In particular, there is reflectometry in the frequency domain, which allows focusing exclusively on vibrations: [https://doi.org/10.1364/OE.27.013923]. Interrogating devices operating in the frequency domain can be implemented on a single photinic chip and are quite lightweight even in comparison with devices for FBGs. 3. Since the experiment must be repeatable, a description of the setup for interrogating the sensor should be provided, with a detailed description of its components and a schematic drawing of the experimental setup or a photograph. If a commercially available setup was used, its model and manufacturer should be provided, as well as the probing and data processing modes.
4. It would also be interesting to know what kind of gratings were used in the study? How contrasting are they?
5. Line 201 says that a physical quantity is measured by measuring wavelength variations. What principle is used to measure the wavelength? A spectrum analyzer with a diffraction grating? A prism projecting light onto a photocell array? Both of these methods do not allow for the rapid measurements needed to detect vibrations. Did you use some kind of filter to convert wavelength changes into intensity changes?
6. The manuscript talks about vibrations, but does not provide a single vibration spectrum. 7. Some of the drawings are located at the end of sections, I would suggest moving them higher, immediately after the first mention in the text.

Reviewer 2 Report

Comments and Suggestions for Authors

    In the present work, with an example of a flexible beam as rocket, the manuscript presented a model-based robust control algorithm by using fiber Bragg grating (FBG) sensor arrays. The experimental results showed that the proposed control strategy can reduce the response of the primary structure significantly. The subject of the paper is of scientific interest and the work presented contributes to the knowledge in the area of vibration control technology. Furthermore, the results presented in this study sound technically correct.

    In general, subject of this manuscript is interesting. This manuscript is in good quality and is suggested to be accepted after the following comments are adequately taken into account.

1. In Figure 1, The photo for FBG strain sensor array which is mounted on the beam should be presented.

2. In Section 2.5, the measured modal response is based on strain mode shapes from the FEA model. If the boundary condition or mass/stiffness of the beam is changed, is it possible obtain the modal response in real-time?

3. From Figure 4, it can be found that there is a time-delay for thrust, does the delay affect on control performance? The authors should give more explanation on this issue.

4. In figs. 8-12, the x-axis unit is second(s), why the response is so slow, please check the unit. According to my understanding, the x-axis unit may be micro-second (ms).

5. In Fig.12, when the step excitation was used, it would be better if the frequency response function could be presented.

Round 2

Reviewer 2 Report

Comments and Suggestions for Authors

The authors have addressed my comments in a satisfactory manner. I suggest to accept the revised document.